# Dual-Responsive Photonic Crystal Sensors Based on Physical Crossing-Linking SF-PNIPAM Dual-Crosslinked Hydrogel

**DOI:** 10.3390/gels8060339

**Published:** 2022-05-30

**Authors:** Wenxiang Zheng, Xiaolu Cai, Dan Yan, Ghulam Murtaza, Zihui Meng, Lili Qiu

**Affiliations:** 1School of Chemistry and Chemical Engineering, Beijing Institute of Technology, Beijing 100081, China; zhengwenxiang2019@163.com (W.Z.); 13091178730@163.com (X.C.); murtaza1405@gmail.com (G.M.); mengzh@bit.edu.cn (Z.M.); 2School of Materials Science and Engineering, Peking University, Beijing 100871, China

**Keywords:** photonic crystals, silk fibroin, physical crossing-linking, biocompatibility, wearable sensing

## Abstract

Flexible wearable materials have frequently been used in drug delivery, healthcare monitoring, and wearable sensors for decades. As a novel type of artificially designed functional material, photonic crystals (PCs) are sensitive to the changes in the external environment and stimuli signals. However, the rigidity of the PCs limits their application in the field of biometric and optical sensors. This study selects silk fibroin (SF) and poly-N-isopropylacrylamide (PNIPAM) as principal components to prepare the hydrogel with the physical crosslinking agent lithium silicate (LMSH) and is then integrated with PCs to obtain the SF-PNIPAM dual-crosslinked nanocomposite for temperature and strain sensing. The structural colors of the PCs change from blue to orange-red by the variation in temperature or strain. The visual temperature-sensing and adhesion properties enable the SF-PNIPAM dual-crosslinked nanocomposite to be directly attached to the skin in order to monitor the real-time dynamic of human temperature. Based on its excellent optical properties and biocompatibility, the SF-PNIPAM dual-crosslinked nanocomposite can be applied to the field of visual biosensing, wearable display devices, and wound dressing materials.

## 1. Introduction

As the new generation of artificially designed optical material, photonic crystals (PCs) have unique periodic structures and optical characteristics—photonic bandgap (PBG)—which can alter the wavelength of reflected light when the stimuli such as pH, temperature, and pressure change the structural color of the PCs [1,2,3,4,5,6,7,8,9,10,11]. Therefore, PCs have been playing an important role in environmental protection, biomimetic, and energy materials, especially in the field of biosensing [12,13,14,15,16,17]. However, the rigidity of the PCs reduces their stability and limits their further application in wearable sensors. Researchers have been striving to combine PCs with other functional materials to improve mechanical properties and further expand their application [18,19]. For example, the nanocomposite materials, which integrate PCs with stimuli-responsive materials and can respond to external stimuli, are used in PC sensors [20]. Recently, the research has mainly focused on integrating PCs with natural materials which are biocompatible, degradable, provide mechanical strengthand eco-friendly, and improving the intensity of the response to external stimuli [17,21,22].

Silk fibroin (SF) is a material with good biocompatibility, breathability, and moisture permeability and has been widely used in the field of biomedicine for slow-release drugs, artificial organs, and immobilized enzyme materials [23,24,25,26,27,28]. Min et al. reported deformable silk-based optical nanostructure, which facilitated the connection between nano-optics and biology [29]. Wang et al. demonstrated photonic crystal heterostructure based on silk fibroin and achieved the controllable adjustment of the spectral response of the material [30].

Here, we prepared a hydrogel interpenetrating polymer network by combining SF with PNIPAM via the physical crosslinking agent lithium silicate (LMSH), which can improve the tensile strength of SF and explore its application. The SF-PNIPAM dual-crosslinked nanocomposite with vivid structural colors was obtained by combining PCs with SF-PNIPAM hydrogel precursor, and the shape of the nanocomposite could be designed according to different molds. Changes in external stimuli such as temperature or strain led to variation in the lattice spacing of PCs resulting in shrinkage or swelling of the SF-PNIPAM dual-crosslinked nanocomposite, along with the change of the structural color. The visual temperature sensing and adhesive properties enabled the SF-PNIPAM dual-crosslinked nanocomposite to be directly attached to the surface of human skin, which also holds great promise in the field of visual wearable temperature sensing; Because of the biocompatibility and water retention, the SF-PNIPAM dual-crosslinked nanocomposite carries the potential to be used as wound dressings materials.

## 2. Results and Discussion

### 2.1. Optimization of the SF-PNIPAM Dual-Crosslinked Nanocomposite

As a hydrogel monomer, NIPAM can shrink with water loss when external temperature changes and will have cliff-like shrinkage near its phase transition temperature (33 °C). However, the poor mechanical properties of materials consisting of pure PNIPAM limit its application in wearable sensors.

We prepared a prepolymer solution by combining SF and PNIPAM to improve their mechanical strength and biocompatibility. The side chain of SF contains a large number of hydrophilic groups such as hydroxyl, carboxyl, and amino groups, the addition of SF would increase the water content of SF-PNIPAM dual-crosslinked nanocomposite. The increase in SF content greatly improved its flexibility and mechanical properties. However, when the content of SF exceeds 20 wt% of NIPAM, the SF-PNIPAM dual-crosslinked nanocomposite would not be formed easily due to the excessive water content and its mechanical properties would be greatly reduced. Moreover, the SF-PNIPAM dual-crosslinked hydrogel hardly combines with the PCs and even damages its structure. Considering the above factors, we added 1 g NIPAM and 0.2 g SF to 10 mL H_2_O, then added LMSH (physical crosslinker), APS (initiator), and TEMED (accelerate the reaction) to prepare the SF-PNIPAM dual-crosslinked nanocomposite, which has best mechanical properties and excellent stability (Appendix A).

The crosslinking principle of hydrogel interpenetrating network is as follows: LMSH adsorbs the monomers (NIPAM) on the surface of its charged nanoparticles, initiates free radical polymerization in situ to form PNIPAM macromolecular chain, meanwhile acts as a physical crosslinking point to adsorb a large number of SF macromolecular chain, finally obtaining dual-crosslinked nanocomposite based on the physical crosslinking (Figure 1) [31,32,33].

### 2.2. The Effect of Crosslinking Temperature on the Properties of the SF-PNIPAM Dual-Crosslinked Nanocomposite

The preparation of SF-PNIPAM dual-crosslinked nanocomposite with NIPAM could be influenced by crossing temperature [34,35,36]. To explore the effect of crosslinking temperature on nanocomposite performance, we prepared SF-PNIPAM dual-crosslinked nanocomposite film at 20 °C and 36 °C, respectively, as shown in Figure 2. The SF-PNIPAM dual-crosslinked nanocomposite film was transparent with good mechanical properties, and the structural color of the PCs was fully displayed at 20 °C. However, the crosslinking temperature (36 °C) exceeds the phase transition temperature of NIPAM (33 °C), the SF-PNIPAM dual-crosslinked nanocomposite film obtained was milky white with poor mechanical properties due to the phase transition, NIPAM changed from homogeneous system to heterogeneous system when it was heated over 33 °C, and its hydrophobic effect is enhanced, which lead to the failure to encapsulate PCs or even destroyed the structure of the PCs, making the structural color fade and disappear. The result illustrates that when crosslinking temperature is higher than the NIPAM’s phase transition temperature, the obtained SF-PNIPAM dual-crosslinked nanocomposite will become milky white, with poor mechanical and elastic properties. Therefore, we chose 20 °C as the crosslinking temperature of the SF-PNIPAM dual-crosslinked nanocomposite.

To observe the microstructure of the SF-PNIPAM dual-crosslinked nanocomposite film, we performed SEM characterization after drying it at room temperature (25 °C), as shown in Figure 3. The structure of the PCs was perfectly preserved during the crosslinking process (Figure 3a,b), and the photonic crystal nanospheres were perfectly wrapped by SF-PNIPAM dual-crosslinked hydrogel. Due to drying at room temperature, the SF-PNIPAM dual-crosslinked nanocomposite film shrunk in volume with the loss of water, and uneven wrinkles appeared on the surface of the film (Figure 3c,d), which could not reflect the real state of its water content.

To observe the real structure of the film in water, we freeze-dried the SF-PNIPAM dual-crosslinked nanocomposite film at −54 °C for 24 h and further characterized by SEM (Figure 4). The freeze-dried SF-PNIPAM dual-crosslinked nanocomposite film presented a honeycomb shape as the constant gel structure and moisture loss; the pore diameter was between 15 μm–20 μm, and the walls of the pore were light and thin (Figure 4a–c), which shows that the SF-PNIPAM dual-crosslinked nanocomposite film had good water retention. Moreover, the 3D PC array was perfectly embedded and wrapped between the hole walls, keeping its structure intact (Figure 4d). Owing to its excellent liquid absorption performance and biocompatibility, the nanocomposite film can be used in the fields of wound adjuvants.

### 2.3. The Response of the SF-PNIPAM Dual-Crosslinked Nanocomposite to Temperature

With the temperature-sensitive monomer—NIPAM—the SF-PNIPAM-dual-crosslinked nanocomposite can respond to temperature, which will shrink/relax with ambient temperature changes. A large extent of sudden contraction/relaxation occurs at the critical phase transition temperature (LCST), which, in turn, leads to a difference in the lattice spacing of the PCs, causing the changes in its PBG and structural color that can be visible to the naked eye. The LCST of pure NIPAM is 33 °C, and the addition of SF increases the LCST of the SF-PNIPAM dual-crosslinked nanocomposite by 2 °C, which is 35 °C. The structural color of the SF-PNIPAM dual-crosslinked nanocomposite will red-shift slightly with temperature decreasing below 35 °C, which indicates less shrinkage. Above 35 °C, cliff-type dehydration shrinkage occurs, and the structural color changes from original blue to red, which is different from normal hydrogel photonic crystal. The thermos-sensitive mechanism of the SF-PNIPAM dual-crosslinked nanocomposite is the interaction between the hydrophobic and hydrophilic groups in the system and water molecules. When the ambient temperature is lower than LCST, the hydrogen bonding between the SF-PNIPAM dual-crosslinked nanocomposite and the water molecules forms a highly ordered in the solvated shell. With the rising temperature, the hydrophobic interaction between molecules is enhanced, and the hydrogen bonds are destroyed. At a critical temperature (LCST), the SF-PNIPAM dual-crosslinked nanocomposite undergoes a phase transition and loses water; its structure changes from a loose coil structure to a compact colloidal structure. The illustration in Figure 5b shows the partial purple and red appearance of the SF-PNIPAM dual-crosslinked nanocomposite at 35 °C. The rapid shrinkage of the film changed the PCs from a close-packed structure to a non-uniform crystal state, resulting in structural color variations. Figure 5a shows the reflection spectra of the SF-PNIPAM dual-crosslinked nanocomposite with different temperatures. Before LCST, the peak position of the nanocomposite film hardly moved and started to move when the temperature reached 31 °C. The peak changed significantly around 35 °C, owing to the obvious shrinkage of the film above the LCST, which caused the periodic close-packed structure of the PCs to be destroyed, and the film varied from its original transparent state to slightly white, so its reflection peak appeared to transform significantly. The results indicate that the LCST of the SF-PNIPAM dual-crosslinked nanocomposite (35 °C) is 2 °C higher than that of NIPAM because the addition of SF increases the LCST of the SF-PNIPAM dual-crosslinked nanocomposite. Since the LCST of the nanocomposite was close to the temperature of the human body, and we can adjust its LCST by adding salt ions and other methods, the SF-PNIPAM dual-crosslinked nanocomposite can be used in the field of the wearable devices to detect body temperature with the naked eye.

### 2.4. The Response of the SF-PNIPAM Dual-Crosslinked Nanocomposite to Pressure

The SF-PNIPAM dual-crosslinked nanocomposite exhibits excellent mechanical properties, which can produce stable deformation, accompanied by the change of photonic crystal structural color, so this material can be used to monitor joint movement and recovery. The structural color changes of the SF-PNIPAM dual-crosslinked nanocomposite are the result of deformation, and external stimuli such as pressure will deform it in the direction perpendicular to the force, which could increase the lattice constant of the PCs, resulting in the red shift of the reflection peak and structural color. The uneven change of structural color is caused by uneven pressure. The shape and structural color of the SF-PNIPAM dual-crosslinked nanocomposite will recover if external stimuli are removed, which indicates that our material has excellent mechanical properties. Figure 6, Figure 7 and Figure 8 show the structural color changes of the SF-PNIPAM dual-crosslinked nanocomposite films with different grades of thickness (1 mm, 2 mm, and 5 mm, respectively) under different deformations. The thickness of the SF-PNIPAM dual-crosslinked nanocomposite films of 1 mm and 2 mm was relatively thin and soft, so the longitudinal deformation during the sensing process was limited and easy to make the structural color uneven (Figure 6 and Figure 7, Appendix A). The original structural color of the SF-PNIPAM dual-crosslinked nanocomposite film with 5 mm was blue (Figure 8), and the structural color gradually changed from blue to blue-green, yellow, and orange with pressure increasing (Appendix A). Removing pressure can make the film immediately returns to its original structural color, and its structural color is more uniform and obvious. Therefore, we chose the SF-PNIPAM dual-crosslinked nanocomposite with a thickness of 5 mm for mechanical sensing.

Due to the existence of a large number of hydrophilic groups, the SF-PNIPAM dual-crosslinked nanocomposite can fit well with the skin without falling off, without employing any adhesive. Figure 9 shows the nanocomposite film attached to the knuckle surface. When the joint was bent, it drove the film to stretch, along with the variation of the PC lattice constant and structural color (Appendix A). Based on the above properties, the SF-PNIPAM dual-crosslinked nanocomposite can be stuck on the surface of the skin, and the change in shrinkage and structural color can indicate the progress of the movement. In addition, because the materials can be attached to joints and will not fall with the movement of the joints, and also can produce visual structural color changes when bending, the SF-PNIPAM dual-crosslinked nanocomposite has the potential application for patient sports rehabilitation monitoring.

### 2.5. The Stability of the SF-PNIPAM Dual-Crosslinked Nanocomposite

In order to explore the acid-base stability of the SF-PNIPAM dual-crosslinked nanocomposite, we prepared a phosphate buffer with pH = 5, 6, 7, 8, 9, and placed the material in the above buffer solution for 10 min. The structural color and reflection peak positions of the SF-PNIPAM dual-crosslinked nanocomposite with different pH environments were essentially unchanged (Appendix A). The reflectance spectrum in Figure 10 indicates that the film had good acid-base stability in buffer solutions at pH 5 and 8, respectively. In weak acid or alkaline environment, a large number of carboxyl groups and amino groups contained in SF can be combined with hydrogen ion and hydroxyl ion in a buffer solution, respectively, which can make the SF-PNIPAM dual-crosslinked nanocomposite to avoid the destruction of the structure by acidic or base environments. The SF-PNIPAM dual-crosslinked nanocomposite has good acid-base stability and tolerance in the above pH range without shrinking or swelling, so the SF-PNIPAM dual-crosslinked nanocomposite has good environmental adaptability.

We characterized the swelling of the SF-PNIPAM dual-crosslinked nanocomposite in water to illustrate the excellent water-absorbing of the nanocomposite. The SF-PNIPAM dual-crosslinked nanocomposite film with the volume of 1 cm × 1 cm × 2 mm was placed in ultrapure H_2_O and measured its swelling of each dimension in 2 h. Figure 11 shows the relationship between the swelling ratio of the film over time, the L_0_, W_0_, and H_0_ were the original size of the SF-PNIPAM dual-crosslinked nanocomposite film, and L, W, and H were the size of the SF-PNIPAM dual-crosslinked nanocomposite film soaked in water for different periods. The result shows that our SF-PNIPAM dual-crosslinked nanocomposite film swelled in all three directions, and the swelling ratio in the height direction was slightly larger than that in the other two directions, which was 1.17. The SF-PNIPAM dual-crosslinked nanocomposite gradually reached swelling equilibrium after 10 h, and its volume remained constant. The addition of SF increased its hydrophilicity, and the film can absorb water and has a strong water-retention performance, which can shrink with water loss and swell again when immersed in water. The SF-PNIPAM dual-crosslinked nanocomposite was placed in the air when its volume shrinks correspondingly with the loss of water, and it can swell again when immersed in a solution, which has the inherent properties of hydrophilic hydrogels. Based on the above properties, the SF-PNIPAM dual-crosslinked nanocomposite film can be used in water-absorbing and drug loading and has the potential application for visual monitoring of healthcare.

## 3. Conclusions

In this paper, we used SF macromolecular from cocoons and PNIPAM macromolecular chains formed by free-radical polymerization of NIPAM monomers, lithium magnesium silicate as a crosslinking agent to prepare a purely physically crosslinked nanocomposite with good biocompatibility, flexibility, and stability. The reaction conditions were mild; the SF-PNIPAM dual-crosslinked nanocomposite obtained that is physically crosslinked has high strength and toughness. Through the optimization of the formula, the SF-PNIPAM dual-crosslinked nanocomposite had the best strength and strain sensing performance when the dosage ratio of NIPAM and SF was 5:1. The SF-PNIPAM dual-crosslinked nanocomposite had temperature sensing performance due to the use of NIPAM, and its phase transition temperature was near 35 °C. The structural color and reflectance spectrum of the PCs were changed with varying temperatures. When subjected to pressure stimulation, the structural color of the SF-PNIPAM dual-crosslinked nanocomposite gradually changes from blue to orange-red, which can realize naked-eye sensing of external strain. At the same time, the excellent acid-based stability of the SF-PNIPAM dual-crosslinked nanocomposite made it suitable in acidic and alkaline environments. Based on optical performance, biocompatibility, adhesion, and acid stability, the SF-PNIPAM dual-crosslinked nanocomposite can be used for body temperature measurement, real-time monitoring of human movement, and other areas.

## 4. Materials and Methods

### 4.1. Materials and Reagents

Methyl methacrylate (MMA, A.R.), potassium persulfate (KPS, A.R.), Alumina (Al_2_O_3_, A.R.); cocoons, sodium carbonate (Na_2_CO_3_, A.R.), lithium bromide (LiBr, 99% A.R.), polyethylene glycol (PEG, 14,000, A.R.), N-isopropylacrylamide (NIPAM, 99% A.R.), n-hexane (A.R.), lithium magnesium silicate ([(Mg, Li)_3_Si_4_O_10_(OH)_2_·4H_2_O], LMSH, A.R.), ammonium persulfate (APS, 98% A.R.), tetramethylethylenediamine (TEMED, 99% A.R.), concentrated sulfuric acid (H_2_SO_4_, 98% A.R.), hydrogen peroxide (H_2_O_2_, 30% A.R.).

### 4.2. Preparation of 3D Photonic Crystals

The polymethylmethacrylate (PMMA) colloidal microspheres were synthesized by soap-free emulsion polymerization [37]. MMA monomer and H_2_O were added to four-necked flask (250 mL), nitrogen was bubbled through the liquid surface for 30 min to remove O_2_, then heated to 80 °C and stirred at 300 rpm. The KPS solution was added to the above mixture and continued to react for 45 min, obtaining PMMA colloidal suspension. The mixture was centrifuged thrice to remove unreacted monomer and initiator. The vertical self-assembly method was used to prepare three-dimensional (3D) photonic crystals; the PMMA colloidal microspheres were self-assembled on a glass slide and obtained 3D photonic crystal array with face-centered cubic (FCC) stacking [38,39].

### 4.3. The NIPAM Recrystallization and Purification

NIPAM (30 g) monomer and toluene (75 mL) were added to a single-necked flask, stirred magnetically, and heated to 45 °C for 30 min to obtain NIPAM solution; then, Al_2_O_3_ (20 g) was immediately added to the above solution, continued stirring for 5 min to remove the polymerization inhibitor in NIPAM, and filtered. n-hexane was added to the solution and placed in the dark at 4 °C for 24 h, and the NIPAM crystals were obtained by suction filtration, dried under vacuum at 40 °C for 24 h, and stored in a sealed and dark place at 4 °C.

### 4.4. Preparation of Silk Fibroin

We prepared a silk protein solution following the protocol [24]. Cocoons were cut and boiled with Na_2_CO_3_ solution (5 g L^−1^) for 30 min. The process was repeated thrice to remove sericin from the silk fibers and dried at room temperature. LiBr solution (9.5 M) was prepared and kept at 60 °C; then, added dried silk fibers with magnetic stirring and continued to react for 4 h. After the silk fibers completely dissolved, dialysis was performed with a dialysis bag (8000–14,000) at 4 °C for three days to remove the LiBr and small protein and obtained SF solution. Then the dialyzed SF solution was concentrated with 10% polyethylene glycol (Mw = 14,000). It was then reverse dialyzed at 4 °C for 12 h to obtain the SF solution with a concentration of about 5 wt%.

### 4.5. Preparation of SF-PNIPAM Dual-Crosslinked Nanocomposite

LMSH (mass ratio 20% of NIPAM), SF (0.2 g), and NIPAM (1 g) were mixed in 9 mL H_2_O and stirred magnetically until fully dissolved, placed in an ice-water mixing bath (<5 °C), and blew N_2_ for 10 min to avoid the presence of oxygen affecting polymerization effect of the prepolymer solution, APS (0.01 g) was dissolved in H_2_O (2 mL) and added to the above solution, TEMED (2%, 100 μL) was added as a catalyst, continued to stir for 10 min and obtained prepolymer solution. The prepolymer solution was injected into the PCs to replace air and reacted at room temperature for 48 h after filling. Then the mold was taken out, and the nanocomposite was placed in H_2_O for 48 h to remove unreacted monomers. The SF-PNIPAM dual-crosslinked nanocomposite film was finally obtained.

## Figures and Tables

**Figure 1 gels-08-00339-f001:**
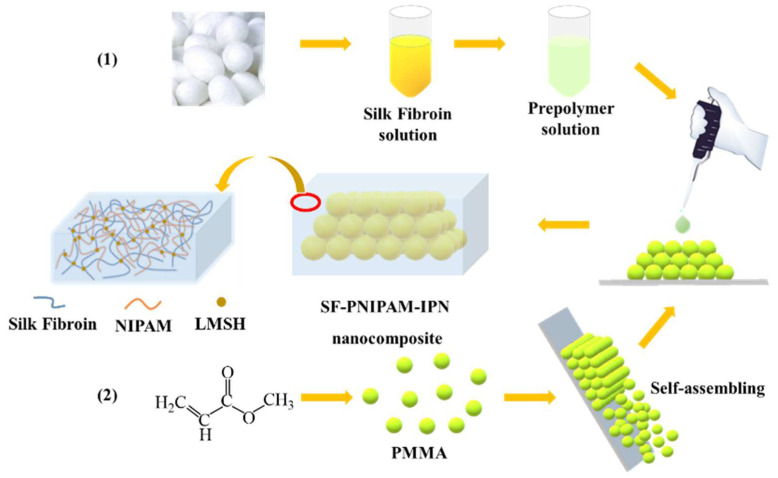
Scheme of crosslinking principle of the SF-PNIPAM dual-crosslinked nanocomposite based on purely physically crosslinking network.

**Figure 2 gels-08-00339-f002:**
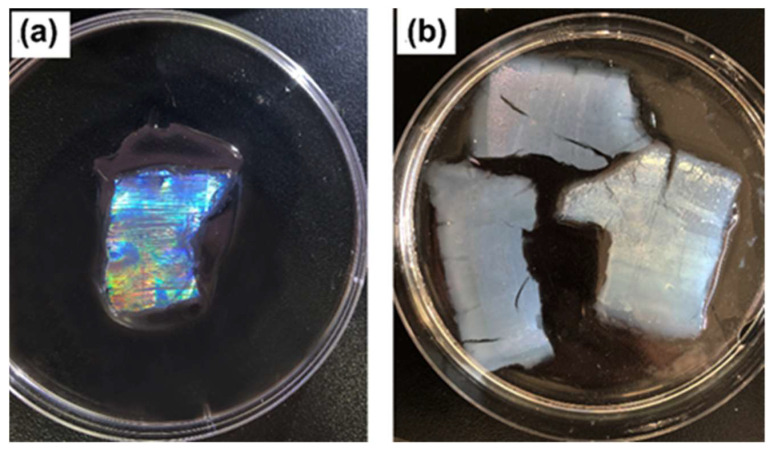
The SF-PNIPAM dual-crosslinked nanocomposite film at different crosslinking temperature (**a**) 20 °C; (**b**) 36 °C.

**Figure 3 gels-08-00339-f003:**
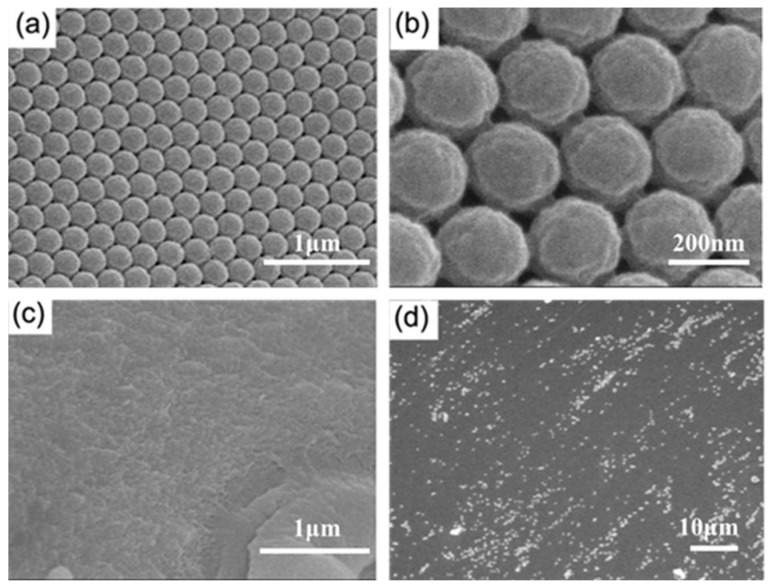
The SEM of the SF-PNIPAM dual-crosslinked nanocomposite film dried at room temperature (**a**) front of the PCs, ×35,000; (**b**) front of the PCs, ×130,000; (**c**) the film surface, ×35,000; (**d**) the film surface, ×1500.

**Figure 4 gels-08-00339-f004:**
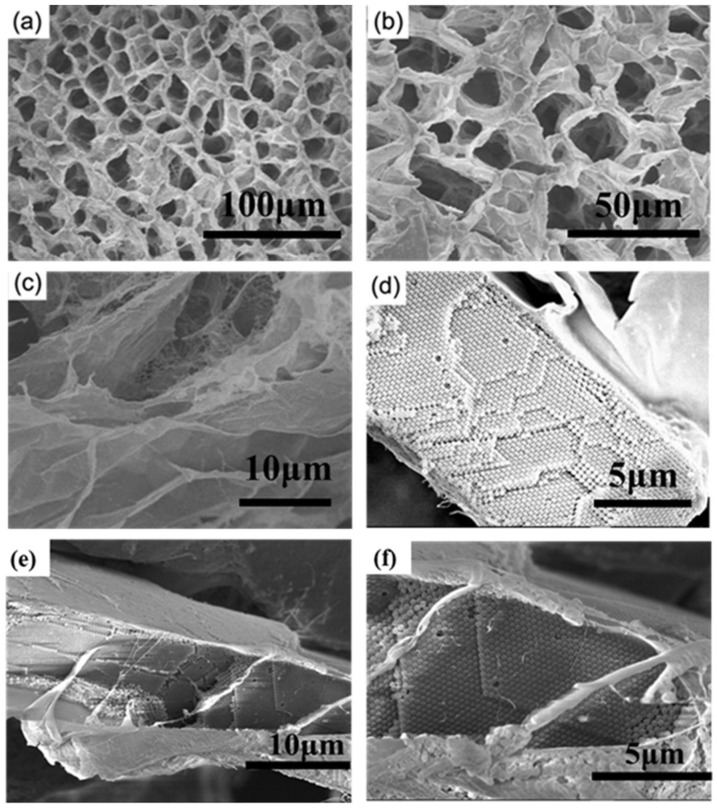
The SEM of freeze-dried the SF-PNIPAM dual-crosslinked nanocomposite film at −54 °C for 24 h (**a**) ×500; (**b**) ×1000; (**c**) ×3300; (**d**) ×7000; cross-section images (**e**) ×3700; (**f**) ×8000.

**Figure 5 gels-08-00339-f005:**
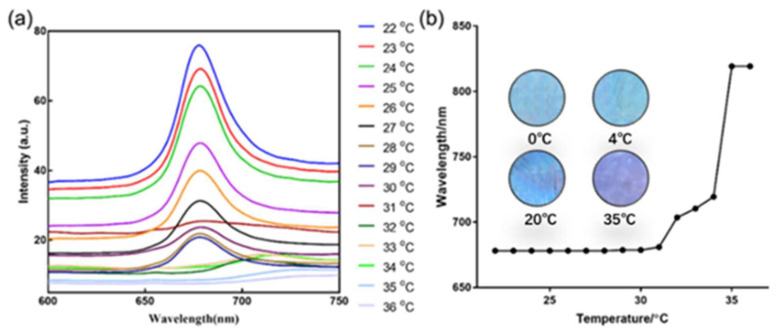
(**a**) The reflection spectrum of the SF-PNIPAM dual-crosslinked nanocomposite films with the change of temperature; (**b**) The relationship between reflection peak wavelength and temperature (Illustration: Structural color of the SF-PNIPAM dual-crosslinked nanocomposite film at a different temperature).

**Figure 6 gels-08-00339-f006:**
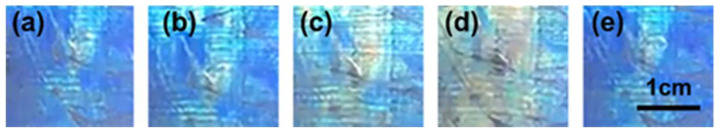
The structural color of the SF-PNIPAM dual-crosslinked nanocomposite film (1 mm) with different deformations (**a**) 0%; (**b**) 5%; (**c**) 10%; (**d**) 15%; (**e**) Remove external force.

**Figure 7 gels-08-00339-f007:**
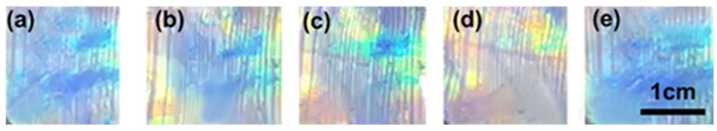
The structural color of the SF-PNIPAM dual-crosslinked nanocomposite film (2 mm) with different deformations (**a**) 0%; (**b**) 5%; (**c**) 10%; (**d**) 15%; (**e**) Remove external force.

**Figure 8 gels-08-00339-f008:**
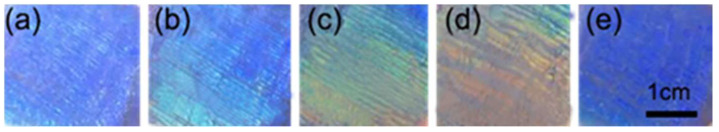
The structural color of the SF-PNIPAM dual-crosslinked nanocomposite film (5 mm) with different deformations (**a**) 0%; (**b**) 5%; (**c**) 10%; (**d**) 15%; (**e**) Remove external force.

**Figure 9 gels-08-00339-f009:**
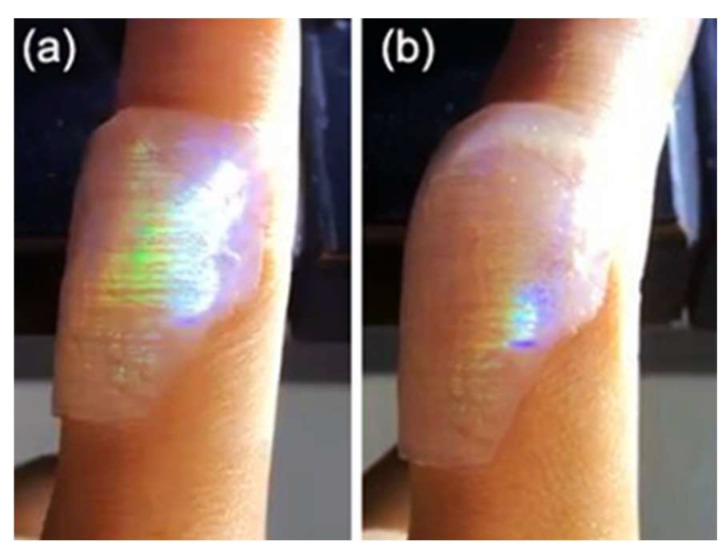
The picture of the SF-PNIPAM dual-crosslinked nanocomposite film attached to the knuckle surface (**a**) past on finger joints; (**b**) bent to 160°.

**Figure 10 gels-08-00339-f010:**
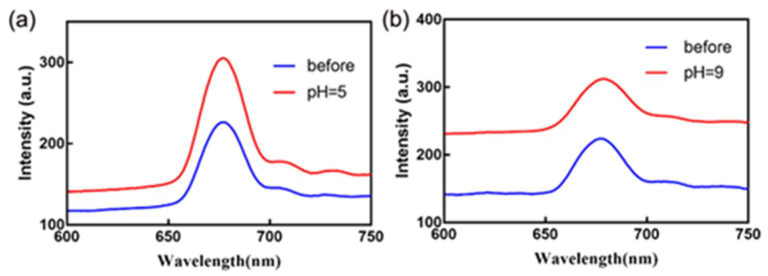
The reflection spectrum of the SF-PNIPAM dual-crosslinked nanocomposite in buffer solution (**a**) pH = 5; (**b**) pH = 9 (blue and red are the reflectance spectrum before and after the film was put into the acid-base buffer, respectively).

**Figure 11 gels-08-00339-f011:**
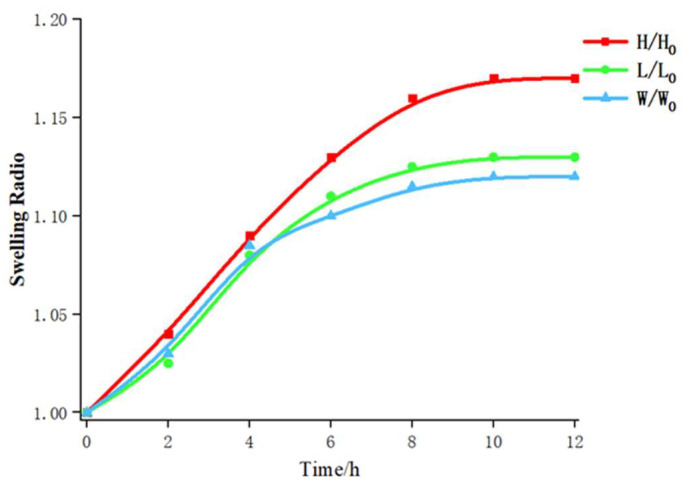
The swelling properties of the SF-PNIPAM dual-crosslinked nanocomposite film.

## Data Availability

The data presented in this study are available on request from the corresponding author.

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
