# Peer review of "Dual-Responsive Photonic Crystal Sensors Based on Physical Crossing-Linking SF-PNIPAM Dual-Crosslinked Hydrogel"

_gels, 2022, doi:10.3390/gels8060339_

Round 1

Reviewer 1 Report

The manuscript describes the synthesis of SF-PNIPAM-IPN nanocomposite and its characterization. The response of the nanocomposite to temperature, strain, pH has been demonstrated with possible explanation. Authors have focused that based on the optical performance, biocompatibility, acid stability, the SF-PNIPAM-IPN nanocomposite can be used for body temperature measurement, real-time monitoring of human movement, drug release monitoring and other areas. Based on careful scrutiny, I recommend the rejection of the manuscript in its present form. A major revision is required before its consideration.

Comments:

  1. The English of the manuscript is very poor. The manuscript should be rewritten and submitted for further review. This is very essential. Many places are difficult to read and understand.
  2. In the description of the work, avoid the abbreviation as best possible. Reader will face a problem to realization the story.
  3. Fig. 1 is difficult to have the reaction concept. Give the presentation in chemical formula also.
  4. Give the scale bar in Fig. 3. The same is true for Fig. 4 also.
  5. A schematic presentation in the introduction part is required to focus the work to the readers.

Author Response

Dear Reviewer:

Thank you very much for reviewing our manuscript “Dual-Responsive Photonic Crystal Sensors Based on Physical Crossing-Linking SF-PNIPAM-IPN Hydrogel” (Manuscript ID: gels-1701920). Your valuable comments and suggestions greatly helped us to improve our manuscript. We have carefully revised our manuscript according to your suggestions, and point-by-point answered the questions.

Thank you and best regards.

Yours sincerely
Lili Qiu

Reviewer 2 Report

  1. The manuscript should be studied again in order to amend the present typos.
  2. In the introduction section, there is no part reviewing similar previous works and the novelty of the present manuscript. It should be added.
  3. The formatting should be changed to match with the Gels journal. For example, the materials and methods section should be after the introduction, not the last part.
  4. English and writing style should be improved significantly before publication in any journal.

Author Response

Dear Reviewer:

Thank you very much for reviewing our manuscript “Dual-Responsive Photonic Crystal Sensors Based on Physical Crossing-Linking SF-PNIPAM-IPN Hydrogel” (Manuscript ID: gels-1701920). Your valuable comments and suggestions greatly helped us to improve our manuscript. We have carefully revised our manuscript according to your suggestions, and point-by-point answered the questions. Please see the attachment.

Kind regards,

Lili Qiu

Reviewer 3 Report

The work "Dual-Responsive Photonic Crystal Sensors Based on Physical 2
Crossing-Linking SF-PNIPAM-IPN Hydrogel" by Zheng et al., is well-written and well-organized.

This work may be accepted without any further reviews.

But,

All the figures must be reworked.

Mainly, the scheme of the work is not showing any scientific details.

Cross-section images may be added.

Fig 6 and & are very tiny.

In fig. 10, what is "Before"?

Author Response

(The authors gave the same response as above.)

Reviewer 4 Report

The work entitled "Dual-responsive photonic crystal sensors based on physical cross-linking SF-PNIPAM-IPN hydrogel" describes the synthesis and properties of a novel type of functional materials. Such a material shows a difference in its color by changing the pressure or the temperature. Although these results are rather interesting, this work is incomplete. The figures are too small, and the English writing is confusing. Some comments, questions, and corrections are given below:

  • The title is an essential part of the manuscript. It should give an idea of the work. Thus, it is crucial to avoid abbreviations. I would recommend changing SF to silk-fibroin.
  • The English writing is confusing. Please check it carefully. For instance:
    1. The sentence on lines 13 to 16, 37 to 41, 51 to 54, and so on.
    2. Avoid using terms like "can't," "didn't," "isn't," etc.
    3. The units must follow a pattern: mL instead of ml.
  • Line 33: "Researchers are committed to combining (…)". Who are they? There is not any reference. Was it combined in a conference? Please, consider changing this statement.
  • Line 70: "SF exceeded that 20% of NIPAM (…)". Is this percentage related to weight or mol?
  • Regarding formula 7, I recommend mentioning its composition in section 2.1.
  • The authors claim that when the content of SF exceeds 20% of NIPAM, it does not form the SF-PNIPAM-IPN nanocomposite due to the excessive water content. Then, they have chosen the formula 7.
    1. Why did they not test the formulas 1 to 3?
    2. What is the difference when the final material is obtained by formulas 6 and 8?
    3. Chain density is an important variable that must be considered. A composition of 0.17g SF:0.85g NIPAM (corresponding to 20%) could lead to a hydrogel with desired properties?
  • Line 97: "effect of amido." What is the meaning of amido?
  • Line 108: "after drying it at room temperature." The room temperature depends on the season and the location of the authors' country. I recommend specifying the temperature value.
  • “Applications in sustained release of drugs” (line 22): The authors have done an essay to confirm this statement?
  • Line 124: "Owing to its excellent liquid absorption performance and biocompatibility, the nanocomposite film can be used in the fields of drug sustained release and wound adjuvants." The authors did not perform swelling measurements, cytotoxicity essays, or even in vitro drug release to support this statement. Thus, this declaration is speculative.
  • Line 166: "The SF-PNIPAM-IPN nanocomposite exhibits excellent mechanical properties". It seems this statement was based on naked-eye observations. The authors did not perform stress-strain measurements. Are you comparing it with another sample or material?
  • The videos are fascinating. However, it is possible to see that only changing the finger position is enough to change the color of the hydrogel. I recommend recording a new video without moving the finger and perhaps using some support of something else to see only the contraction effect.
  • The capital letters L, H, and W are related to length, height, and width? Please indicate it in the manuscript.
  • Line 230: "The addition of SF increased its hydrophilicity". The authors did not evaluate the effect of SF content in hydrogel swelling to support this statement.
  • Line 232: "The SF-PNIPAM-IPN nanocomposite was placed in the air when its volume shrinks correspondingly with the loss of water, and it can swell again when immerse in solution". The authors claim this effect is related to SF content. However, this property is not necessarily related to SF content; it is intrinsic to hydrogels which must have hydrophilic polymers in their constitution.
  • Is your hydrogel an IPN, semi-IPN, or physically cross-linked?

Author Response

(The authors gave the same response as above.)

Round 2

Reviewer 1 Report

The manuscript has been updated and there is still option to change the English. Try to do once before final submission. The manuscript can be recommended for publication. 

Author Response

Dear Reviewer:

Thank you very much for reviewing our manuscript “Dual-Responsive Photonic Crystal Sensors Based on Physical Crossing-Linking SF-PNIPAM Dual-crosslinked Hydrogel” (Manuscript ID: gels-1701920) again. Your valuable comments and suggestions greatly helped us to improve our manuscript. We have carefully checked our manuscript according to your suggestions, and point-by-point answered the questions as following.

  1. “Flexible wearable materials have frequently appeared in the application fields of drug delivery, health detection, and wearable sensing research for decades.” has been replaced by “Flexible wearable materials have frequently been used in drug delivery, healthcare monitoring, and wearable sensors for decades.” (Line 10).
  2. “artificial” has been replaced by “artificially” (Line 11).
  3. “the changes in external environment and the stimulus-signals” has been replaced by “the changes in external environment and the stimuli- signals” (Line 12).
  4. “the application” has been replaced by “their application”, (Line 13).
  5. “This paper” has been replaced by “This study” (Line 13).
  6. “hydrogel solution with physical cross-linking agent lithium silicate (LMSH), then combines integrated with PCs” has been replaced by “the hydrogel with physical cross-linking agent lithium silicate (LMSH), and is then integrated with PCs” (Line14 to 15).
  7. “mechanical detection” has been replaced by “strain sensing” (Line 16).
  8. “The structural colors of the PCs to change from blue to orange-red by the various of the temperature or strain” has been replaced by “The structural colors of the PCs change from blue to orange-red by the variation in temperature or strain” (Line 17).
  9. “characteristics” has been replaced by “properties” (Line 18).
  10. “on the skin in order to monitor real-time dynamic of human temperature.” has been replaced by “to the skin in order to monitor the real-time dynamic of human temperature.” (Line 18 to 19).
  11. “characteristics” has been replaced by “properties” (Line 19).
  12. “wearable display devices, wound dressing and drug sustained release materials.” has been replaced by “wearable display devices and wound dressing materials” (Line 21).
  13. “new generation” has been replaced “the new generation”, “artificial” has been replaced “artificially” (Line 25).
  14. “reflection light when the stimulus changes the structural color of the PCs, such as pH, temperature, pressure and other stimuli [1-11].” has been replaced “reflected light when the stimuli such as pH, temperature, and pressure change the structural color of the PCs [1-11].” (Line 27 to 28).
  15. “the PCs has” has been replaced by “PCs have” (Line 28).
  16. “its” has been replaced by “their” (Line 30).
  17. “its” has been replaced by “their”, “sensing research” has been replaced by “sensors.” (Line 31).
  18. “committed to combining the PCs” has been replaced by “been striving to combine PCs” (Line 32).
  19. “its application fields” has been replaced by “their application” (Line 33).
  20. “the PCs nanocomposite materials, which combined the PCs with stimulus responsive materials, can respond to external stimuli and used in field of the PCs sensors” has been replaced by “the PCs nanocomposite materials, which integrate PCs with stimuli-responsive materials, can respond to external stimuli are used in PCs sensors” (Line 33 to 35).
  21. “the research of the PCs has mainly focused on combining with natural materials with biocompatible, degradable, and ecofriendly, which can improve the effect of the re-sponse to external stimuli and researching on the biocompatibility and mechanical strength of the PCs” has been replaced by “the research has mainly focused on integrating PCs with natural materials which are biocompatible, degradable, provide mechanical strength are eco-friendly, and can improve the intensity of the response to external stimuli.” (Line 35 to 37).
  22. “Silk Fibroin (SF), with good biocompatibility” has been replaced by “Silk Fibroin (SF) is a material with good biocompatibility” (Line 38).
  23. “has been widely used in the field of biomedicine, such as drug slow-release agents,” has been replaced by “and has been widely used in the field of biomedicine, for drug slow-release drugs,” (Line 39).
  24. “which” has been added in Line 41.
  25. “achieve” has been replaced by “achieved the” (Line 43).
  26. “We prepared hydrogel interpenetrating polymer network with purely physical crossing-linking by combining SF with PNIPAM via physical cross-linking agent lithium silicate (LMSH)” has been replaced by “Here, we prepared a hydrogel interpenetrating polymer network by combining SF with PNIPAM via the physical cross-linking agent lithium silicate (LMSH)” (Line 45 to 46).
  27. “was” has been added in Line 48.
  28. “and the shape of nanocomposite can be” has been replaced by “and the shape of the nanocomposite could be” (Line 49).
  29. “will cause the lattice distant of PCs varies with the shrinkage” has been replaced by “led to variation in the lattice spacing of PCs resulting in shrinkage” (Line 51).
  30. “enable” has been replaced by “enabled” (Line 53).
  31. “good biocompatibility” has been replaced by “the biocompatibility” (Line 55).
  32. “obtains” has been replaced by “carries” (Line 56).
  33. “wound dressings and drug slow release materials.” has been replaced by “wound dressings materials.” (Line 57).
  34. “consisted by pure PNIPAM limits its application in the field of wearable detections.” has been replaced by “consisting of pure PNIPAM limit its application in the wearable sensors.” (Line 62 to 63).
  35. “a” has been added in Line 64.
  36. “performance” has been replaced by “strength” (Line 65).
  37. “SF will increase water content” has been replaced by “SF would increase the water content” (Line 67).
  38. “will greatly improve” has been replaced by “greatly improved” (Line 68).
  39. “exceeded that” has been replaced by “exceeds than” (Line 69).
  40. “its mechanical properties will become greatly reduced.” has been replaced by “and its mechanical properties would greatly reduce. (Line 70).
  41. “combine” has been replaced by “combines” (Line 72).
  42. “damage” has been replaced by “damages” (Line 72).
  43. “obtained interpenetrating polymer network” has been replaced by “obtaining dual-crosslinked nanocomposite” (Line 81).
  44. “Since the use of NIPAM, the preparation of SF-PNIPAM-IPN nanocomposite can be influenced by crossing-temperature” has been replaced by “The preparation of SF-PNIPAM dual-crosslinked nanocomposite with NIPAM could be influenced by crossing-temperature” (Line 88).
  45. “the” has been added in Line 94.
  46. “change occurs” has been replaced by “transition” (Line 96).
  47. “will changing” has been replaced by “changed” (Line 96).
  48. “it” has been added in Line 97.
  49. “weaken” has been replaced by “fade” (Line 99).
  50. “the” has been added in Line 101.
  51. “elasticity” has been replaced by “elastic” (Line 102).
  52. “dried” has been replaced by “drying” (Line 111).
  53. “the” has been added in Line 112.
  54. “In order to” has been replaced by “To” (Line 118).
  55. “hour” has been replaced by “hours” (Line 119).
  56. “a” has been added in Line 120.
  57. “was” has been added in Line 121
  58. “the” has been added in Line 122.
  59. “response” has been replaced by “respond” (Line 132).
  60. “of” has been added in Line 133.
  61. “a” has been added in Line 134.
  62. “of” has been replaced by “in” (Line 134).
  63. “of” has been replaced by “in” (Line 135).
  64. “increasing” has been replaced by “decreasing” (Line 138).
  65. “changing” has been replaced by “changes” (Line 139).
  66. “In fact, the thermosensitive mechanism” has been replaced by “The thermo-sensitive mechanism” (Line 140).
  67. “with temperature rising” has been replaced by “With the rising temperature” (Line 144).
  68. “will undergo” has been replaced by “undergoes” (Line 146).
  69. “change” has been replaced by “transition” (Line 147).
  70. “the” has been replaced by “The” (Line 149).
  71. “caused the PCs to change” has been replaced by “changed the PCs” (Line 149).
  72. “temperature” has been replaced by “temperatures” (Line 152).
  73. “its” has been added in Line 155.
  74. “transforming” has been replaced by “transform” (Line 156).
  75. “indicates” has been replaced by “indicate” (Line 156).
  76. “a” has been added in Line 167.
  77. “changing” has been replaced by “changes” (Line 171).
  78. “cause it deform” has been replaced by “deform it” (Line 173).
  79. “the” has been added in Line 174.
  80. “the” has been replaced by “The” (Line 174).
  81. “removed” has been replaced by “is removing” (Line 176).
  82. “shows” has been replaced by “show” (Line 177).
  83. “thicknesses” has been replaced by “grades of thickness” (Line 178).
  84. “with 1 mm and 2 mm was” has been replaced by “of 1 mm and 2 mm was” (Line 180).
  85. “a” has been added in Line 187.
  86. “and employ no adhesive” has been replaced by “without employing any adhesive.” (Line 198).
  87. “the change of shrink” has been replaced by “the change in shrinkage” (Line 202).
  88. “the” has been added in Line 212.
  89. “a” has been added in Line 218.
  90. “the” has been added in Line 219.
  91. “shrink or swell” has been replaced by “shrinking or swelling” (Line 221).
  92. “the” has been added in Line 229.
  93. “different periods of time” has been replaced by “different periods of time” (Line 233).
  94. “direction” has been replaced by “directions” (Line 235).
  95. “remains” has been replaced by “remained” (Line 237).
  96. “water retention” has been replaced by “water-retention” (Line 238).
  97. “losing” has been replaced by “loss” (Line 239).
  98. “immerse” has been replaced by “immersed” (Line 241).
  99. “have” has been replaced by “has” (Line 241).
  100. “made use of” “used” (Line 248).
  101. “a” has been added in Line 250.
  102. “the” has been added in Line 254.
  103. “phase inversion temperature was at near 35 °C” has been replaced by “phase transition temperature was near 35 °C” (Line 256).
  104. “will be” has been replaced by “were” (Line 257).
  105. “Added MMA monomer and H2O in four-necked flask (250 mL), blew nitrogen under the liquid surface” has been replaced by “MMA monomer and H2O were added to a four-necked flask (250 mL), nitrogen was bubbled through the liquid surface” (Line 276 to 277).
  106. “then” has been added in Line 278.
  107. “obtained” has been replaced by “obtaining” (Line 279).
  108. “Centrifugal washing 3 times” has been replaced by “The mixture was centrifuged thrice” (Line 280).
  109. “a” has been added in Line 282.
  110. “Added NIPAM (30 g) monomer and toluene(75mL) into a single-necked flask” has been replaced by “NIPAM (30 g) monomer and toluene(75mL) were added to a single-necked flask” (Line 285).
  111. “immediately added Al2O3(20 g) into the above solution” has been replaced by “then Al2O3(20 g) was immediately added to the above solution” (Line 286).
  112. “Added n-hexane” has been replaced by “n-hexane was added” (Line 288).
  113. “According to previous research, we prepared silk protein solution” has been replaced by “We prepared a silk protein solution following the protocol” (Line 293).
  114. “Cut cocoons and boiled them” has been replaced by “Cocoons were cut and boiled” (Line 293).
  115. “repeated 3 times” has been replaced by “The process was repeated thrice” (Line 294).
  116. “and” has been added in Line 295.
  117. “Prepared LiBr solution (9.5 M)” has been replaced by “LiBr solution (9.5 M) was prepared” (Line 295).
  118. “then” has been added in Line 296.
  119. “dialysis with dialysis bag (8000-14000)” has been replaced by “dialysis was performed with a dialysis bag (8000-14000)” (Line 297).
  120. “and” has been added in Line 299.
  121. “Reverse dialyzed” has been replaced by “It was then reverse dialyzed” (Line 300).
  122. “obtained” has been replaced by “obtain” (Line 301).
  123. “a” has been added in Line 301.
  124. “Added LMSH (mass ratio 20 % of NIPAM), SF (0.2g), and NIPAM (1g)” has been replaced by “LMSH (mass ratio 20 % of NIPAM), SF (0.2g), and NIPAM (1g) were mixed” (Line 303).
  125. “an” has been added in Line 304.
  126. “dissolved APS (0.01 g) in H2O (2 mL) and added to the above solution, then added TEMED (2 %, 100 μL) as catalyst, continued to stirring” has been replaced by “APS (0.01 g) was dissolved in H2O (2 mL) and added to the above solution, TEMED (2 %, 100 μL) was added as a catalyst, continued to stir” (Line 306 to 307).
  127. “filling completely” has been replaced by “filling” (Line 309).
  128. “swelled” has been replaced by “placed” (Line 310).
  129. “the SF-PNIPAM-IPN nanocomposite film was obtained finally” has been replaced by “The SF-PNIPAM dual-crosslinked nanocomposite film was finally obtained” (Line 311).

Reviewer 2 Report

There are still typos and grammatical mistakes that make the manuscript very hard to follow. It is required that the authors ask a native person to help them improve the text.

Author Response

(The authors gave the same response as above.)

Reviewer 4 Report

The work entitled "Dual-responsive photonic crystal sensors based on physical cross-linking SF-PNIPAM-IPN hydrogel" describes the synthesis and properties of a novel type of functional materials. Such a material shows a difference in its color by changing the pressure or the temperature.

The authors have made considerable efforts to correct the English language and answer the questions. The manuscript has been improved, demonstrating the quality of such work. However, there are still some corrections and misconceptions that they must correct before acceptance. Thus, I recommend the approval of this manuscript with minor revisions. They are listed below:

  1. Line 37: "mate-rials." Please, change it to "materials."

  1. In my previous question regarding the drug release applications, I mentioned that claiming their materials are suitable for sustainable drug release without performing any experiment sounds speculative. They argued swelling measurements support their statement. In addition, they declare the application in this declaration is “potential”.

Unfortunately, there are many phases throughout the manuscript highlighting that their hydrogels can be used for sustained drug release (lines: 21, 57-59, and 127). The word "potential application" refers to results obtained after performing In vitro tests. This statement can confuse the readers expecting to see some experiments and results regarding drug release without finding it. Moreover, the swelling measurements are insufficient to determine if their hydrogel is suitable for drug release.

Much of the current literature on drug release shows that the swelling/deswelling mechanism is entirely different from drug release (or its absorption). Thus, the water absorption essay is not enough to describe the drug release. In fact, swelling and drug release experiments may complement each other.

There is a significant difference in the conceptual meaning between "drug release", "controlled drug release", and "sustained controlled drug release". The first term is often used for describing a simple release of a drug. The second one relates to controlling/decreasing the burst release effect and controlling the release of drugs.

Finally, the third one is related to controlling the drug release maintaining preferred drug concentrations in the blood within an identified therapeutic window.

For instance:

a. Zamani, F. et al. Chapter 7 - Nanofibrous and nanoparticle materials as drug-delivery systems. In nanostructures for drug delivery (2017), 239-270

b. Lee, JH. et al. Controlled Drug Release from Pharmaceutical Nanocarriers. Chemical engineering science (2015), 125, 75-84.

c. Raja, MM. et al. Chapter 18 - Polymeric Nanomaterials: Methods of Preparation and Characterization. In: Nanocarriers for Drug Delivery (2019), 557-653.

An important parameter in drug release is the partition coefficient which is the ratio of the concentration of a substance in one medium or phase (C1) to the concentration in a second phase (C2). The partition coefficient between the hydrogel and drug is specific. Even if a hydrogel absorbs a drug, there is no assurance whether the drug will be released without performing experiments. Thus, you must inform which drug you are using in this experiment.

In my opinion, the characterization of hydrogels was well done for the desired application (sensor). The inclusion of other speculative applications does not improve the quality of this work. Thus, based on this reasoning, I strongly recommend removing all sentences related to the drug release in the manuscript.

  1. Section 2.2 (Lines: 98-99): The authors described the effect of temperature on the properties of hydrogel nanocomposite. They mentioned a predominance of the isopropyl's hydrophobic effect over the amide's hydrophilic effect.

The LCST is related to a dynamic conformational change in the PNIPAM chains. Thus, such property depends on a set of molecules and is unrelated to a single effect of amine or isopropyl groups.

As demonstrated by Alaghemandi M. et al., this effect is not observed in a single chain system, contradicting the author's statement.

Thus, I recommend correcting this sentence.

For instance:

a. Alaghemandi, M. et al. Macromol. Theory Simul. (2012), 21, 106-112.

b. Ahiabu, A and Serpe, M. J. ACS Omega (2017), 2, 1769-1777.

c. Qui, Y. et al. - Ch. 4 Improving mechanical properties of injectable polymers and composites (2011), 61-91. In: Injectable Biomaterials. Science and Applications.

d. Akiyama, T. et al. - Ch. 9 Temperature-responsive polymers for cell culture and tissue engineering applications (2015), 203-233. In: Switchable and Responsive Surfaces and Materials for Biomedical Applications.

  1. I have questioned the authors if their hydrogel was IPN or semi-IPN. They responded, "Our hydrogel is physically cross-linked by PNIPAM and SF absorbed on the LMSH, at the same time, PNIPAM and SF formed an IPN network, so the hydrogel of this work based on physical crossing-linking Silk Fibroin -PNIPAM-IPN Hydrogel."

However, IUPAC defines IPN as: "A polymer comprising two or more networks which are at least partially interlaced on a molecular scale but not covalently bonded to each other and cannot be separated unless chemical bonds are broken. A mixture of two or more pre-formed polymer networks is not an IPN".

According to this definition and the author's response, their hydrogels are neither IPN nor semi-IPN. The same network consisting of PNIPAM, SR, and LMSH has both the chemical and the physical cross-linking. Then, they obtained a dual-crosslinked hydrogel. Such classification is in line with their materials and methods section.

Based on the above considerations, I recommend changing "IPN" to "dual-crosslinked hydrogel".

For instance:

a. Zoratto, N. and Matricardi, P. Chapter 4-Semi-IPNs and IPN-based hydrogels. In: Polymeric Gels (2018), 91-124.

b. Zhang, X. et al. New Journal of Chemistry (2020), 44, 9903-9911

c. Wei, S. et al. International Journal of Biological Macromolecules (2020), 155, 153-162.

Author Response

Dear Reviewer:

Thank you very much for reviewing our manuscript “Dual-Responsive Photonic Crystal Sensors Based on Physical Crossing-Linking SF-PNIPAM Dual-crosslinked Hydrogel” (Manuscript ID: gels-1701920) again. Your valuable comments and suggestions greatly helped us to improve our manuscript. We have carefully checked our manuscript according to your suggestions, and point-by-point answered the questions as following.

Point 1: Line 37: "mate-rials." Please, change it to "materials."

Response 1

Thank you for your great suggestions. We have changed “mate-rials” to “materials”.

Point 2: In my opinion, the characterization of hydrogels was well done for the desired application (sensor). The inclusion of other speculative applications does not improve the quality of this work. Thus, based on this reasoning, I strongly recommend removing all sentences related to the drug release in the manuscript.

Response 2

Thank you for your great suggestions. We discussed and decided to delete the relevant content about drug release. (lines: 21,57-59, 123, 194, 255)

Point 3: Section 2.2 (Lines: 98-99): The authors described the effect of temperature on the properties of hydrogel nanocomposite. They mentioned a predominance of the isopropyl's hydrophobic effect over the amide's hydrophilic effect.

The LCST is related to a dynamic conformational change in the PNIPAM chains. Thus, such property depends on a set of molecules and is unrelated to a single effect of amine or isopropyl groups.

Thus, I recommend correcting this sentence.

Response 3

Thank you for your great suggestions. “NIPAM changed from homogeneous system to heterogeneous system when it was heated over 33℃ , and the hydrophobic effect of the isopropyl on the macromolecular chain was stronger than the hydrophilic effect of the amide, which lead to the failure to encapsulate PCs or even destroyed the structure of the PCs , making the structural color fade and disappear.” has been replaced by “NIPAM changed from homogeneous system to heterogeneous system when it was heated over 33℃ , and its hydrophobic effect is enhanced, which lead to the failure to encapsulate PCs or even destroyed the structure of the PCs , making the structural color fade and disappear ”.

Point 4: I recommend changing "IPN" to "dual-crosslinked hydrogel".

Response 4

Thank you for your great suggestions. Based on your careful considerations, we have changed "IPN" to "dual-crosslinked hydrogel".